# Multi-Algorithm Analysis Reveals Pyroptosis-Linked Genes as Pancreatic Cancer Biomarkers

**DOI:** 10.3390/cancers16020372

**Published:** 2024-01-15

**Authors:** Kangtao Wang, Shanshan Han, Li Liu, Lian Zhao, Ingrid Herr

**Affiliations:** 1Department of General, Visceral & Transplant Surgery, Molecular OncoSurgery, Section Surgical Research, University of Heidelberg, 69117 Heidelberg, Germany; shanshan.han@stud.uni-heidelberg.de (S.H.); l.liu@uni-heidelberg.de (L.L.); lian.zhao@stud.uni-heidelberg.de (L.Z.); i.herr@uni-heidelberg.de (I.H.); 2Department of General Surgery, The Xiangya Hospital, Central South University, Changsha 410008, China

**Keywords:** pancreatic cancer, bibliometric analysis, LDA analyses, machine learning, pyroptosis

## Abstract

**Simple Summary:**

Pancreatic ductal adenocarcinoma (PDAC) is often diagnosed at advanced stages, resulting in limited treatment options and poor survival rates. To address this challenge, we conducted a comprehensive analysis of pyroptosis-related genes using advanced algorithms. Our study, involving 1273 PDAC cases, identified 357 pyroptosis-related genes. Notably, BHLHE40, IL18, BIRC3, and APOL1 were found to be related to unfavourable PDAC outcomes and were validated through experiments and multiple datasets. We developed a novel model and an accessible nomogram to predict PDAC prognosis. Our research enhances our understanding of PDAC and has significant implications for both research and clinical practice.

**Abstract:**

Pancreatic ductal adenocarcinoma (PDAC) is often diagnosed at late stages, limiting treatment options and survival rates. Pyroptosis-related gene signatures hold promise as PDAC prognostic markers, but limited gene pools and small sample sizes hinder their utility. We aimed to enhance PDAC prognosis with a comprehensive multi-algorithm analysis. Using R, we employed natural language processing and latent Dirichlet allocation on PubMed publications to identify pyroptosis-related genes. We collected PDAC transcriptome data (n = 1273) from various databases, conducted a meta-analysis, and performed differential gene expression analysis on tumour and non-cancerous tissues. Cox and LASSO algorithms were used for survival modelling, resulting in a pyroptosis-related gene expression-based prognostic index. Laboratory and external validations were conducted. Bibliometric analysis revealed that pyroptosis publications focus on signalling pathways, disease correlation, and prognosis. We identified 357 pyroptosis-related genes, validating the significance of BHLHE40, IL18, BIRC3, and APOL1. Elevated expression of these genes strongly correlated with poor PDAC prognosis and guided treatment strategies. Our accessible nomogram model aids in PDAC prognosis and treatment decisions. We established an improved gene signature for pyroptosis-related genes, offering a novel model and nomogram for enhanced PDAC prognosis.

## 1. Introduction

Pancreatic ductal adenocarcinoma (PDAC) is one of the deadliest cancers, often diagnosed at an advanced metastatic stage [1,2,3]. Despite significant advancements in treatment modalities, such as surgery, chemotherapy, and targeted therapies, the 5-year overall survival rate for PDAC patients in 2022 remained dishearteningly low at 11% [4]. This bleak prognosis primarily arises from 90% of PDAC cases being identified in advanced stages, extending beyond the pancreas and spreading systemically, with over 50% developing metastases [4]. This dire situation underscores the urgent need for new biomarkers to assist in PDAC risk assessment and to identify novel therapeutic targets.

Recent studies have shed light on the detrimental effects of first- and second-line chemotherapeutic drugs, which induce pyroptosis and exacerbate the progression and chemoresistance of PDAC [5]. Consequently, pyroptosis-related gene signatures have emerged as valuable tools for predicting prognosis in PDAC [6,7,8,9]. Pyroptosis, a relatively newfound form of inflammatory caspase-induced lytic programmed cell death, is notably activated in infected cells to eliminate pathogenic niches, incite inflammation, and attract immune cells [10]. Characterised by cell swelling, membrane rupture, the release of cellular contents, and initiation of a potent inflammatory response [11,12], pyroptosis is initiated by inflammatory caspases-1/4/5/11, which cleave and activate gasdermin D (GSDMD) to execute the process. The active N-terminal fragment of GSDMD binds to membrane lipids, disrupting membrane integrity and forming pores, ultimately causing changes in cell osmotic pressure, cellular swelling, and membrane rupture [13]. This cascade of events includes the maturation and secretion of numerous proinflammatory cytokines like IL-18 and IL-1β, which activate active and passive immunity, fostering a robust inflammatory response [14].

Pyroptosis plays a pivotal role in the immune response against infections, especially viral ones like HIV and COVID-19 [15,16]. However, its role in cancer development and treatment is complex, influenced by factors such as tumour heterogeneity, biological behaviours, and epigenetic characteristics [17]. Studies have shown that pyroptosis can contribute to the inflammatory microenvironment of tumours, promoting tumour cell growth and invasion [14]. Conversely, pyroptosis has been demonstrated to activate the immune response and enhance the effectiveness of immunotherapy. Moreover, various chemotherapy agents, such as decitabine (DAC), iron oxide, and glucose oxidase, have shown promise in inducing pyroptosis in cancer cells, thus triggering antitumour immunotherapeutic responses [17]. Nevertheless, our understanding of the function of pyroptosis in PDAC, particularly its impact on prognosis, remains limited. In this study, we harnessed natural language processing (NLP) to analyse the pyroptosis-related literature within the PubMed database comprehensively.

Our investigation focused on identifying research hotspots and extracting pertinent genes related to pyroptosis. Subsequently, we performed a meta-analysis using publicly available PDAC sequencing data, explicitly targeting pivotal genes involved in pyroptosis. Building upon this analysis, we developed a prognostic risk model based on the upregulation of four leading candidate pyroptosis-related genes to predict PDAC prognosis more accurately. Ultimately, our research underscores the significant role of heightened pyroptosis in PDAC prognosis.

## 2. Materials and Methods

### 2.1. Retrieval and Downloading of Pyroptosis-Related Publications

To access publications related to “Pyroptosis”, we employed the pubquery package in R (version: 4.2.1). This package facilitated the retrieval and downloading of pertinent PubMed publications. A comprehensive record of search results in XML format, spanning publications until 31 December 2022, was obtained. Excel (Microsoft Corporation, Redmond, WA, USA) and R were primarily used for visualisation.

### 2.2. Natural Language Processing (NLP) and Latent Dirichlet Allocation (LDA)

To extract detailed publication data, including publication year, region, abstract, and research type, the Python programming language (version 3.11.1), known for its efficiency in object-oriented programming, was employed. The LDA technique was utilised to discern research topics covered in the publications. For this analysis, we set the number of identified topics to 50, considering factors such as appropriate perplexity, redundancy, and legibility.

Using the LDA algorithm, we computed topic probabilities for each article and assigned a topic to each publication based on these probabilities. Heatmaps were generated to visually represent research topics and publication dates [18,19]. For cluster analysis and the creation of thematic networks to discern relationships between themes, we employed the Louvain algorithm within Gephi software (version 0.9.2).

To establish connections between themes, we identified the two topics with the highest attribution probability in each article and counted their co-occurrences within each document. The codes used for the LDA analysis are described (Appendix A).

### 2.3. Patient Clinical Information, Transcriptome Data, and Immunohistochemistry Acquisition

The RNA-sequencing data (FPKM values) and corresponding clinical data from PDAC patients were sourced from the following databases: TCGA database (n = 177; https://www.cancer.gov/tcga (accessed on 9 January 2024)), ICGC data portal including the PACA-AU project for PDAC (n = 91; https://dcc.icgc.org/projects/PACA-AU (accessed on 9 January 2024)), ICGC-PACA-CA (n = 213; https://dcc.icgc.org/projects/PACA-CA (accessed on 9 January 2024)), GEO database (https://www.ncbi.nlm.nih.gov/geo/ (accessed on 9 January 2024)) with the datasets GSE85916 (n = 79), GSE71729 (n = 105), GSE62452 (n = 66), GSE57495 (n = 63), and GSE21501 (n = 192), and E-MTAB database (n = 287; https://www.ebi.ac.uk/biostudies/arrayexpress/studies/E-MTAB-6134 (accessed on 9 January 2024)), totalling 1273 patients. Data normalisation was performed using the log2 transformation, and analysis was conducted using R and R Bioconductor software packages (R version number 4.1.1., R Bioconductor is an open source package) (Appendix A contains all included datasets). Immunohistochemistry (IHC) staining of PDAC patient tissues was obtained from the Human Protein Atlas (TCIA) database (https://www.proteinatlas.org/ (accessed on 9 January 2024)) [20].

### 2.4. Retrieval and Acquisition of Pyroptosis-Related Genes

To compile a comprehensive list of pyroptotic-related genes, the Genes and Expression section of the NIH National Library of Medicine (https://www.ncbi.nlm.nih.gov/guide/genes-expression/ (accessed on 9 January 2024)) was utilised to identify gene symbols. Abstracts of relevant publications were retrieved. All genes and gene names appearing in these abstracts were extracted, and their frequency of occurrence was recorded. Additional verification was conducted to ensure data integrity and establish a correlation between the genes and pyroptosis, referencing databases such as GSEA (https://www.gsea-msigdb.org/gsea/index.jsp (accessed on 9 January 2024)), Genecards (https://www.genecards.org/ (accessed on 9 January 2024)), and KEGG (https://www.genome.jp/kegg/ (accessed on 9 January 2024)). Inclusion criteria: Publications must have abstracts containing gene names or gene symbols. Exclusion criteria: Our researchers manually examined each identified gene, requiring explicit evidence in papers demonstrating the gene’s association with pyroptosis. This association should originate from (1) molecular biology experiments, including but not limited to RT-qPCR at the transcriptional level and protein-level detection represented by Western blot. (2) Sequencing data suggesting correlation. (3) At least two authors agreed that this gene is related to pyroptosis. A union of pyroptosis-related genes was created by consolidating information from all these databases.

### 2.5. Meta-Analysis of Prognostic Implications of Pyroptosis-Related Core Genes in PDAC

To assess the overall survival (OS) implications of pyroptosis core genes in PDAC, we computed the hazard ratio (HR) for each gene using the log-rank test in R. If no significant heterogeneity was observed (I^2^ < 50% and *p* > 0.05), we pooled the HRs of each pyroptosis gene from different bulk sequencing-based cohorts using a fixed-effects model. The Meta package in R (https://cran.r-project.org/web/packages/meta/index.html (accessed on 9 January 2024)) was used for conducting the meta-analysis, and Graphpad (https://www.graphpad.com/, Version 9, (accessed on 9 January 2024)) was used for visualisation. The meta-analysis included all transcriptional data mentioned above in point Section 2.3.

### 2.6. Identification of Differentially Expressed Pyroptosis-Related Genes

We retrieved pyroptosis-related genes, as described above in point Section 2.3, resulting in 357 genes for analysis. Four different datasets were selected: GSE15471 (adjacent tissue = 36, tumour tissue = 36), GSE62452 (adjacent tissue = 61, tumour tissue = 69), GSE71729 (adjacent tissue = 46, tumour tissue = 145), and GSE102238 (adjacent tissue = 50, tumour tissue = 50). These datasets included 193 adjacent tissue samples and 300 tumour tissue samples. To identify differentially expressed genes (DEGs), we used the “limma” package in R software version 4.1.2, applying the criteria of *p*-value < 0.05 and |log2FC| > 0.58. The final set of differentially expressed genes was obtained through intersection analysis.

### 2.7. Cell Culture

Established human PDAC cell lines, including MIA-PaCa2 (RRID: CVCL_0428), BxPc-3 (RRID: CVCL_0186), PANC-1 (RRID: CVCL_0480), and AsPC-1 (RRID: CVCL_0152), and the non-malignant pancreatic ductal cell line CRL-4023 (RRID: CVCL_C466) were obtained from the American Type Culture Collection (ATCC, Manassas, VA, USA). PDAC cells were cultured at 37 °C in high-glucose DMEM (Sigma, Deisenhoffen, Germany), 10% FBS (Sigma), and 25 mmol/L HEPES (Thermo Fisher, Dreieich, Germany). CRL-4023 cells were cultured in a glucose-free mixture of 75% DMEM, 2 mM L-glutamine, 1.5 g/L sodium bicarbonate, and 25% M3 basal medium (Incell Corporation LLC, San Antonio, TX, USA). Monthly testing using PlasmoTest™ (InvivoGen, San Diego, CA, USA) confirmed the absence of mycoplasma contamination in these cell lines. Additionally, all cell lines underwent recent validation through single-nucleotide polymorphism (SNP) analysis conducted by Multiplexion (Heidelberg, Germany).

### 2.8. mRNA Extraction and RT-qPCR

The RNeasy Mini Kit (QIAGEN, Hilden, Germany) was used for mRNA extraction. Following the instructions of the manufacturer, reverse transcription was performed using the High-Capacity RNA-to-DNA™ Kit (Thermo Fisher Scientific, Dreieich, Germany). RT-qPCR was conducted using the PowerUp™ SYBR™ Green Master Mix (Thermo Fisher Scientific, Germany). Primer sequences are available (Appendix A) and were synthesised by Eurofinsgenomic (Ebersberg, Germany). The primer concentration used was 500 nM. Gene expression levels were normalised to the housekeeping gene glycerinaldehyde-3-phosphate-dehydrogenase (GAPDH). The qPCR conditions involved denaturation at 95 °C for 15 s, annealing at 60 °C for 15 s, and extension at 72 °C for 1 min, repeated for 40 cycles. The results are expressed as relative expression values, calculated using the 2-ΔΔCt method [21].

### 2.9. Identification of Key Prognostic Genes and Establishment of a Scoring System for Pyroptosis-Related Genes Prognostic Index

We performed a univariate Cox regression analysis to assess the survival significance of pyroptosis-related genes, considering a significance threshold of *p* < 0.05. Subsequently, we employed the least absolute shrinkage and selection operator (LASSO) regression analysis to narrow the selection of candidate genes. The LASSO regression helped determine the optimal penalty parameter (λ) based on the minimum parameter. Subsequently, we calculated the regression coefficients of these selected pyroptosis-related genes, and a multivariate Cox regression analysis was performed to construct a scoring system. The following formula represents the scoring system:$$Risk\;Score=\sum_{k=1}^{n}coef\left({Gene}^{k}\right)*expr\left({Gene}^{k}\right)$$

In this formula, each gene is denoted as Gene^k^, where k represents the gene index. Coef (Gene^k^) corresponds to the regression coefficient of Gene^k^ obtained from the multivariate Cox regression analysis, while expr(Gene^k^) represents the expression level of Gene^k^. The risk score for an individual is calculated by summing the product of each gene’s coefficient and expression level.

### 2.10. Validation of the Pyroptosis-Related Genes Prognostic Index Scoring System

We conducted a series of analyses to validate the effectiveness of our prognostic index scoring system based on pyroptosis-related genes. First, we generated receiver operating characteristic (ROC) curves for 1-year, 3-year, and 5-year survival using the “survivalROC” package (Version: 1.0.3.1). By calculating the corresponding area under the curve (AUC), we assessed the predictive accuracy of our scoring system. Furthermore, we categorised all patients into low-risk and high-risk groups using the optimal cut-off value of the risk score obtained from the “extra value” package. We utilised Kaplan–Meier survival curves to confirm the prognostic difference between these two groups. Considering the importance of external validation for prognostic features, we employed the ICGC-PACA-CA datasets (available at https://dcc.icgc.org/projects/PACA-CA (accessed on 9 January 2024)) to validate the prognostic value of our scoring system.

### 2.11. Evaluation of Predictive Value and Construction of a Nomogram Prediction Model

To further assess our risk groupings’ predictive value, we performed univariate and multivariate Cox regression analyses using the TCGA and ICGC-CA datasets. These analyses aimed to determine the association between risk groupings and the prognosis of PDAC patients. Additionally, we constructed nomograms for the TCGA and ICGC-CA datasets using the “rms” R package (open source package). These nomograms were designed to predict the survival probabilities of individuals with PDAC at 1, 3, and 5 years. Calibration curves were created to assess the accuracy of the nomogram predictions.

### 2.12. Statistical Analysis

The RT-qPCR data are presented as mean values and standard deviations from a minimum of three independent experiments. Statistical analysis was performed using R Studio (version: 2023.12.0+369) and Excel (version: 2019). A significance level of *p* < 0.05 was considered statistically significant. In GSEA, a false discovery rate (FDR) of 5% was adjusted to account for multiple tests. All *p*-values were calculated based on two-sided statistical tests, with results with a *p*-value of less than 0.05 considered statistically significant. Significance levels are * *p* < 0.05, ** *p* < 0.01, and *** *p* < 0.001.

## 3. Results

### 3.1. Study Design

We employed data mining and NLP to analyse pyroptosis-related literature from the PubMed database (Figure 1). Our study aimed to identify key research trends, hotspots, and relevant genes to pyroptosis. In the discovery phase, we undertook a meta-analysis using public PDAC sequencing data, focusing on genes closely related to pyroptosis. In the training phase, we built a prognostic risk model based on the upregulation of four major pyroptosis-related genes to refine PDAC prognosis prediction. In the validation phase, we tested our model on new data to gauge its accuracy. We confirmed the model’s effectiveness in predicting PDAC prognosis using tests like ROC and survival curves.

### 3.2. Identification of Pyroptosis-Relateted Genes through Bibliometric Analysis

We undertook a bibliometric analysis to pinpoint manuscripts discussing genes related to pyroptosis. Our assessment encompassed 4970 publications up to 31 December 2022. Through evaluating the relationship between publication year (x) and number of publications (y), we discerned that the function y = 1.615e^0.39x^ could represent the data. This exponential correlation suggests a swift surge in publication volume. By the close of 2022, publications tallied up to 2584, but this figure is projected to ascend to 3810 by the culmination of 2023, showcasing a marked upward trend (Figure 2A).

Based on our LDA results, we earmarked prevalent themes in current pyroptosis research. The primary topic was “Signal pathways research”, closely followed by “Disease-related research” and “Risk and prognosis research” (Figure 2B). Within “Signalling Pathways Research”, key areas included “Inflammasome”, “Caspase”, and “GSDME and Therapy”. In the “Disease-Related Research” domain, pivotal subjects included “IL Expression and Regulation”, “Reperfusion Injury”, and “Pyroptosis Model”. The “Risk and Prognosis Research” dimension covered chronic illnesses, tumours, and infections. Moreover, there is considerable exploration around the roles of “GSDMD” and “lncRNA” in pyroptosis.

Current research focal points comprise “Signal Pathways Research”, “Inflammasome”, “Caspase”, “IL Expression and Regulation”, and “Apoptosis and Ferroptosis”, and these trends are expected to continue in the coming years (Figure 2C). Additionally, “Metabolism”, “Inflammasome”, and “Genetics” emerged as the triad of most recurrently broached subjects in pyroptosis-centric articles (Figure 2D). The search algorithm spotlighted NLRP3 as the paramount pyroptosis-linked gene, followed by GSDMD, NLRP1, AIM2, TLR2, GSDME, STAT3, XIST, NLRC4, and HMGB1 (Figure 2E). Upon manual review of the literature, we recognised 357 pyroptosis-relevant genes, embracing both the conventional GSDM and caspase lineages, as well as numerous atypical pyroptosis-related genes (compare Appendix A).

### 3.3. Transcriptome Meta-Analysis Suggests an Essential Role of Pyroptosis Genes in PDAC Prognosis

To underscore the role of pyroptosis signalling in PDAC, we carried out ssGSEA and found a notable downregulation of the pyroptosis signalling pathway in tumour samples compared to adjacent non-tumour tissues. The normalised enrichment score was −1.851, supported by a false discovery rate (FDR) q-value under 0.01 (Figure 3A). To delve deeper into the prognostic implications of the key genes driving pyroptosis—specifically CASP1, CASP3, CASP4, CASP5, GSDMA, GSDMB, GSDMC, GSDMD, and GSDME—we analysed eight PDAC transcriptome datasets, totalling 1273 PDAC patient samples (Figure 3B). A meta-analysis of these datasets indicated that higher CASP1 expression correlated with a 12% decrease in death risk for PDAC patients, with HR = 0.88, 95% CI 0.80–0.96 (Figure 3C). In contrast, elevated expressions of GSDMC with HR = 1.13, 95% CI 1.01–1.28 (Figure 3D), and GSDME with HR = 1.15, 95% CI 1.01–1.28 (Figure 3E), correlated to a 13% and 15% risk in PDAC-related mortality, respectively. For other pyroptotic genes, no statistically significant prognostic relationships were found. In essence, our data emphasise the pronounced impact of certain pyroptotic genes on the prognosis of PDAC patients.

### 3.4. Differential Expression of Key Pyroptosis-Related Genes in PDAC

To identify key pyroptosis-related genes crucial to PDAC prognosis, we analysed four datasets: GSE15471 (adjacent tissue = 36, tumour tissue = 36), GSE62452 (adjacent tissue = 61, tumour tissue = 69), GSE71729 (adjacent tissue = 46, tumour tissue = 145), and GSE102238 (adjacent tissue = 50, tumour tissue = 50). Our differential gene expression analysis, represented by heatmaps (Figure 4A) and volcano plots (Figure 4B), shows differential gene expression. For instance, in GSE15471, 1968 genes exhibited higher expression in tumour tissues, while 775 genes showed lower expression in paracancerous tissues. Comparable patterns were observed across the other datasets. From the 357 pyroptotic genes, we identified 11 consistently expressed genes across all datasets, as shown by a Venn diagram (Figure 4C). Notably, TRIM31 (logFC = 1.14), ANXA1 (logFC = 1.05), GBP1 (logFC = 1.01), BIRC3 (logFC = 0.99), APOL1 (logFC = 0.97), IL18 (logFC = 0.94), ANXA2 (logFC = 0.89), BHLHE40 (logFC = 0.83), and EPHA2 (logFC = 0.75) exhibited significant upregulation in tumour tissues. In contrast, TCEA3 (logFC = −0.99) and BNIP3 (logFC = −1.46) demonstrated higher expression in paracancerous tissues (Figure 4D).

Furthermore, we evaluated mRNA levels of these genes in the established PDAC cell lines MIA-PaCa2, BxPc-3, PANC-1, and AsPC-1 using RT-qPCR. The non-malignant pancreatic ductal cell line CRL-4023 served as a control. Notably, BHLHE40, IL18, BIRC3, and APOL1 had elevated expression in PDAC cells (Figure 5A). However, other genes could not be expressed consistently in multiple datasets in PDAC cell lines, and TRIM31 was not expressed in any group of cell lines (Figure 5B). Immunohistochemical staining from the Human Atlas Protein database revealed increased protein expression of BHLHE40, IL18, BIRC3, and APOL1 in PDAC patient tissues (Figure 5C,D). Specifically, BHLHE40 was expressed in 73% of tumour tissues, while IL18 was universally present in 100% of tumours. BIRC3 was detected in 50% of tumours, and APOL1 expression in 64%. Notably, these proteins displayed minimal expression in non-malignant pancreatic tissues. These findings underscore the potential of these pyroptosis-related genes as prognostic biomarkers and therapeutic targets for PDAC.

### 3.5. Prognostic Scoring System Based on Pyroptosis-Related Genes for PDAC

We conducted a univariate Cox analysis to gauge the prognostic value of these four candidate genes for PDAC. The results pinpointed BHLHE40 (*p* < 0.001, HR = 1.77, 95% CI, 1.02–1.31), BIRC3 (*p* < 0.001, HR = 1.41, 95% CI, 1.17–1.70), and APOL1 (*p* = 0.001, HR = 1.29, 95% CI, 1.11–1.50) as significant risk factors (Figure 6A,B). Leveraging LASSO regression, we constructed a risk model incorporating all four genes (Figure 6C). The risk score was calculated using the following formula:$$Risk\;Score=0.380*expr\left(IL18\right)+0.111*expr\left(BHLHE40\right)+0.100*expr\left(BIRC3\right)+0.012*expr\left(APOL1\right)$$

Harnessing this pyroptosis-focused risk model, we divided TCGA and ICGC patient data into low-risk and high-risk groups, accounting for 50% each (Figure 6D,E). Principal component analysis (PCA) and t-distributed stochastic neighbor embedding (tSNE) analyses underlined a pronounced demarcation between these groups, reflecting notable variations in disease signatures. ROC analysis vouched for the system’s sterling classification prowess (Figure 6F,G), making the risk divisions evident.

Kaplan–Meier survival curves flagged a substantially compromised survival rate for the high-risk faction across both TCGA and ICGC sets (Figure 6H,I). Specifically, high-scoring TCGA patients bore a median survival span of 42.58 ± 20.74, whereas their low-scoring counterparts had a median survival span of 61.50 ± 3.41 (*p* = 0.003). For the ICGC set, the survival was 38.00 ± 5.16 for the high-scoring group and 56.08 ± 5.69 for the low-scoring one (*p* = 0.025). Together, this pyroptosis gene-centric scoring system holds promise as a valuable prognostic tool for categorising PDAC patients, potentially guiding clinical determinations.

### 3.6. Construction and Evaluation of a Prognostic Nomogram Based on Core Pyroptosis Gene Expression

To evaluate the clinical significance of our scoring system, we ran a univariate analysis on PDAC patients using the TCGA database. This encompassed age, gender, tumour grade, N stage, T stage, and our specific score. The study yielded an HR of 2.823 (95% CI, 1.653–4.823) for our risk score, highlighting its significance with a *p*-value of <0.001 (Figure 7A). This suggests that individuals in the high-risk category face almost three times (2.8-fold) the mortality risk compared to those in the low-risk group. Expanding our examination, we carried out a multivariate analysis factoring in patient age, tumour grade, and risk score. This confirmed the high-risk group as a substantial risk determinant for PDAC, evidenced by an HR of 2.77 (1.59–4.82) and a *p*-value < 0.001 (Figure 7B). A heatmap underscores the strong association between our score and diverse clinical determinants. We then crafted a nomogram integrating these clinical parameters to offer a precise prognosis evaluation (Figure 7C,D). To evaluate the predictive precision of our model, we reviewed the calibration curve across both the TCGA training set and the ICGC database, which reflected commendable accuracy. Our scoring model is open to the public and primed for clinical application, promoting its effortless incorporation into clinical decision-making processes. The compelling evidence from our findings implies that this scoring paradigm is a robust prognostic instrument for PDAC. Our scoring tool is available at: https://nomogram-uniheidelberg.shinyapps.io/DynNomapp/ (accessed on 9 January 2024), Figure 7E,F. See (Appendix A) for examples.

## 4. Discussion

This study aimed to investigate the potential of pyroptosis-related gene expression as a prognostic signature for treatment decision making in PDAC. We employed NLP and LDA to screen approximately 5000 pyroptosis-related publications, thereby identifying key research areas in the field. Through search algorithms, we identified 357 genes related to pyroptosis. Meta-analysis confirmed a strong correlation between the expression of pyroptosis effector genes and PDAC prognosis. We analysed multiple transcriptome datasets to explore further the differential expression of these 357 genes in PDAC tumours and adjacent tissues. Subsequently, utilising various algorithms, we developed a PDAC prognosis assessment model based on the expression levels of four key pyroptosis-related genes. This model exhibited excellent predictive capabilities for PDAC prognosis. Notably, this study represents the first comprehensive analysis for identification of key genes among known pyroptosis-related genes using text analysis and subsequent analysis in PDAC. We have rigorously assessed the nomogram model and made it publicly available online for interested users.

To comprehensively identify genes related to pyroptosis, we followed a two-step approach. First, we downloaded all publications related to pyroptosis and used manual gene retrieval methods and multiple databases. This process resulted in a total of 357 pyroptosis-related genes. We aimed to maximise the confirmation of these genes and explore their expression and functional relevance in PDAC. Previous studies that constructed a pyroptosis-related gene signature in PDAC, such as the ones conducted by Huang et al., 2023 [6], Li et al., 2022 [7], Song et al., 2022 [8], Zhu et al., 2023 [9], and Wang et al., 2023 [22], primarily used the literature and reviews to identify candidate genes, confirming around 33 pyroptosis-related genes and subsequent investigations and modelling. Regardless of the approach used for model construction, the unquestionable impact of pyroptosis-related genes on the prognosis of PDAC remains evident. Our approach allowed us to adopt a more comprehensive perspective to avoid missing potential candidate genes and ensure knowledge completeness.

Additionally, we employed bibliometric analysis to assess the research landscape of pyroptosis, utilising LDA topic modelling from machine learning to identify significant research focuses. We found that the research on pyroptosis is still in its early stages, with significant attention directed towards “Signalling”, followed by “Disease-related” and “Risk and prognosis”. Furthermore, we discovered substantial gaps in pyroptosis-related research concerning drug development and clinical applications. Our study introduces a novel algorithm that offers a deeper research perspective than traditional bibliometric analysis of the pyroptosis literature [23,24]. Using new NLP tools and large language models (LLMs) like ChatGPT, our research represents a small branch of the relevant studies and potentially provides a more detailed approach to furthering the field [25].

Our study deviated from typical modelling approaches by initially keeping the data separate and giving priority to experimental validation. Many studies encounter difficulties validating their findings after using mathematical or computational models to analyse and understand a particular subject or problem. For instance, Zhao et al., 2022 [26], merged multiple datasets to identify a lactic acid metabolism-related gene signature in lung adenocarcinoma. In contrast, our study followed a straightforward approach, conducting differential analysis for each gene across four PDAC GEO datasets. To minimise batch effects, we refrained from merging the datasets and instead performed differential analysis individually for each gene in the four PDAC GEO datasets before intersecting the results. Dvinge et al., 2014 [27], highlighted that even rigorous studies using the TCGA database might mask tumour characteristics due to variations in sample processing among different research institutions, emphasising the importance of investigating the diversity between normal and tumour cells. Therefore, we advanced the validation of expression differences between normal cells and tumour-related genes and employed protein analysis from multiple standard PDAC cell lines and large-scale databases.

Despite conducting hundreds of transcriptome sequencing and cross-validation across multiple batches, our RT-qPCR data could not entirely obtain consistent data, which may be because of PDAC cell line composition and tumour heterogeneity. One possible explanation for this discrepancy is that the established cell lines we validated consist solely of PDAC cells, while the transcriptomic data originated from PDAC tumour tissue from patients. Spatial transcriptomic studies by Ma et al., 2022 [23], suggested that PDAC, a highly heterogeneous tumour, comprises a mixture of tumour cells, inflammatory cells, fibrotic tissue, and normal pancreatic tissue. The proportion of PDAC cells can vary significantly, ranging from 10% to 90%, while sequencing captures the entire tissue, resulting in quantitative discrepancies [25,28]. On the other hand, our validation process successfully identified four genes that exhibited high expression in established PDAC cell lines, and we further confirmed their protein expression levels using the Human Atlas database. This enables precise localisation and establishes a solid foundation for future research endeavours.

We have identified four key pyroptosis clinical roles for the first time, but the specific mechanism of action still needs to be further explored. For instance, BHLHE40 (Basic helix-loop-helix family member e40), also known as DEC1 or HLHB2, has been found to control circadian rhythm and cell differentiation [29,30,31,32]. Single-cell sequencing data obtained by Wang et al., 2023 [31], revealed that BHLHE40-driven pro-tumour neutrophils exhibit hyperactivated glycolysis in the pancreatic tumour microenvironment, promoting adverse outcomes in PDAC. BIRC3 (baculoviral iAp repeat containing 3), a member of the family of inhibitors of apoptosis proteins (IAPs), regulates cell death and survival [33,34]. It possesses both anti-apoptotic and pro-pyroptotic functions, promoting cell survival and protecting against pyroptosis while triggering cell death through activation of caspase-1 [33]. BIRC3 is highly expressed in PDAC and may contribute to cancer progression by modulating cell survival and death [35,36]. However, further research is necessary to elucidate its precise mechanisms and develop targeted treatment strategies [37].

APOL1, known as apolipoprotein L1, encodes a protein involved in lysosomal degradation and lipid metabolism [38,39]. The primary focus of APOL1 research has been on kidney diseases, particularly its association with focal segmental glomerulosclerosis and chronic kidney disease [40]. Hu et al., 2012 [38], used a mass spectrometry-based pipeline to identify APOL1 as a novel PDAC biomarker. Xu et al., 2023 [41], employed single-cell analysis and machine learning and discovered that elevated APOL1 levels predict PDAC prognosis and endocrine metabolism. Furthermore, Lin et al., 2021 [42], demonstrated that APOL1 could activate the NOTCH1 signalling pathway, promoting PDAC proliferation while inhibiting apoptosis. Our study also observed an HR of 1.29 (1.11–1.50) for APOL1 in PDAC, revealing high expression at both mRNA and protein levels, thus further substantiating the need for extended APOL1 research.

Interleukin-18 (IL-18) is a proinflammatory cytokine implicated in immune response regulation and the pathogenesis of diverse diseases, including cancer and inflammation [43]. This potent cytokine modulates immune responses and inflammation in PDAC [44]. IL-18 promotes cytokine production and stimulates immune cell activation, including T cells and natural killer cells, pivotal in the anticancer immune response [45]. However, our findings indicate a negative prognosis association between IL-18 and PDAC across multiple datasets [46]. Thus, we speculate that while IL-18 influences immune responses, clearance of infections, and repair of damaged cells, its proinflammatory attributes may contribute to disease progression. Numerous gaps remain in pyroptosis research that necessitate further exploration.

The limitations of this study stem from the algorithm’s lack of interpretability, preventing us from further understanding the scoring criteria for evaluating the prognosis of PDAC patients based on the four genes. Furthermore, the interrelationships among these four genes are still unknown. We validated the four candidate genes at both the transcriptomic and protein levels using PDAC cell lines, a factor that may potentially impact the model’s effectiveness. Additionally, our reliance on transcriptomic profiling, as opposed to more advanced next-generation sequencing (NGS) techniques for model construction, represents another limitation. However, despite these constraints, our validation across multiple datasets demonstrated the robust performance and clinical significance of the scoring system. Our research introduces an innovative model capable of identifying crucial genes from a vast body of literature, leveraging extensive transcriptomic data, and employing various machine learning algorithms. We have unveiled a clinically relevant pathway that can guide scientific investigations.

## 5. Conclusions

Using machine learning, we developed a novel model that identifies key genes by analysing vast transcriptomic data. Our result has provided significant insights into the role of pyroptosis-related genes in PDAC prognosis. The identified gene features and our nomogram offer a promising predictive tool for patient outcomes and treatment planning. However, the study has limitations. It relies on public databases and needs further validation in broader clinical contexts. It is crucial to understand how these pyroptosis-related genes affect PDAC progression and their interplay with other pathways. Future work should focus on these genes’ functional roles in PDAC and their potential as therapeutic targets.

## Figures and Tables

**Figure 1 cancers-16-00372-f001:**
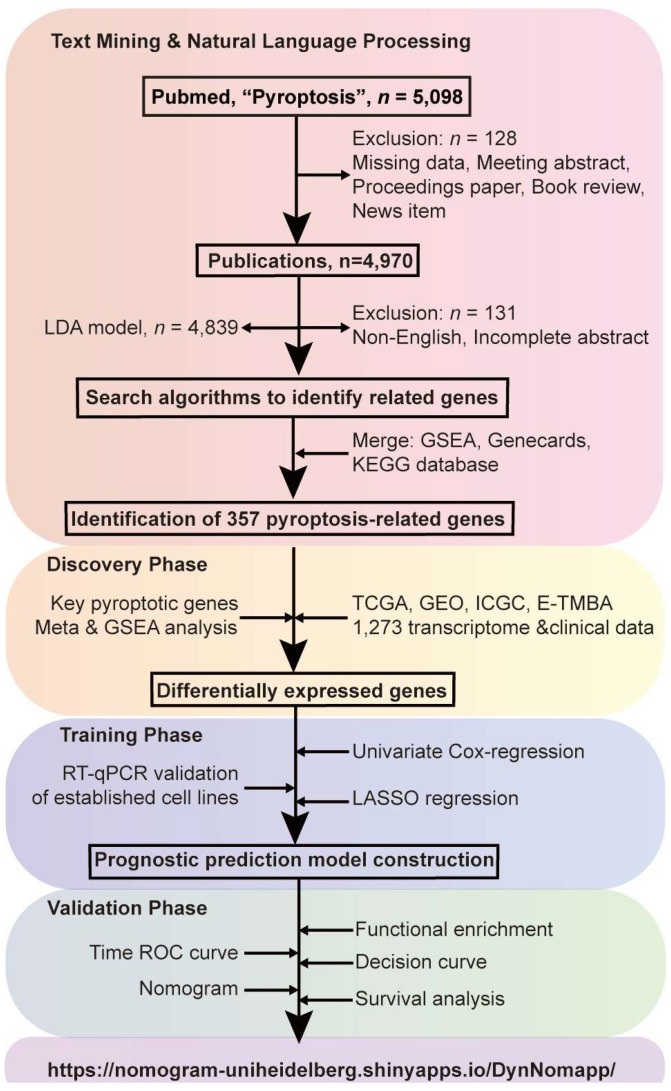
Study design and flowchart. The study unfolds in three phases, delineated by color-coded sections. Pink signifies text mining and natural language processing; using R, we extracted 4970 pyroptosis-related articles from PubMed after thorough filtering. Bibliometric and LDA analyses pinpointed research trends. Yellow highlights model discovery, where meta-analysis and GSEA investigated pyroptosis genes, revealing their significance in PDAC. Blue represents training, encompassing univariate, multivariate, and LASSO regression analyses, leading to a prognostic model. Green represents the validation phase, which utilises methods like ROC analysis to affirm the model’s efficacy. The model is available at https://nomogram-uniheidelberg.shinyapps.io/DynNomapp/ (accessed on 9 January 2024).

**Figure 2 cancers-16-00372-f002:**
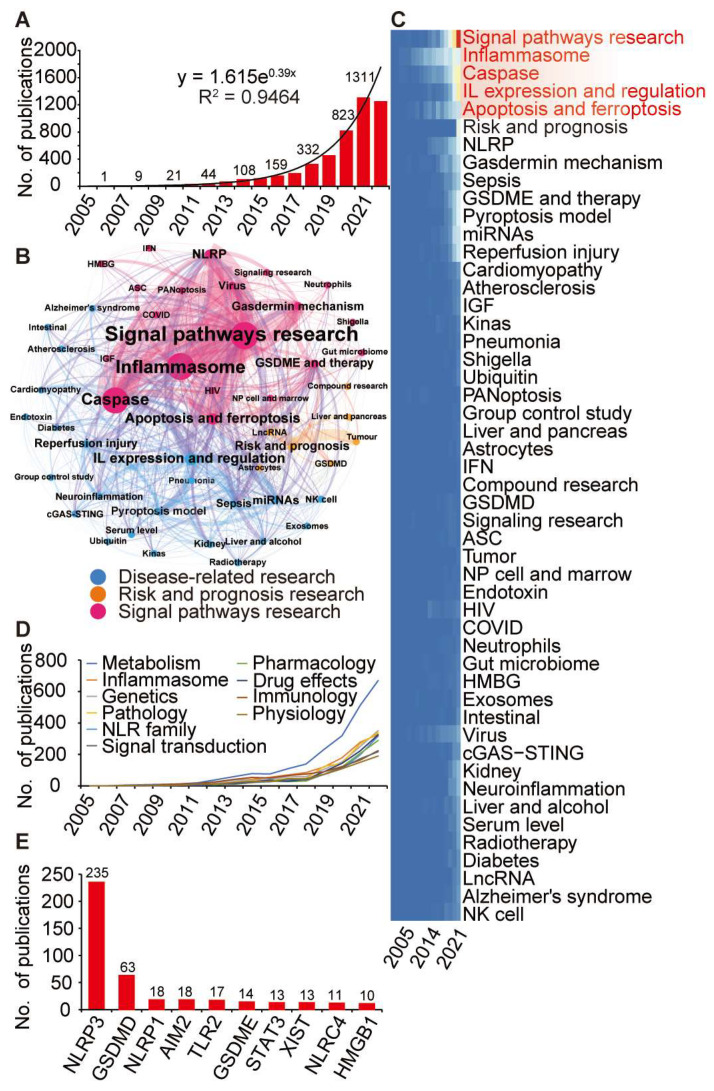
Bibliometric analysis and identification of 357 pyroptosis-related genes. (**A**) The number of publications on pyroptosis and the fitting curves. (**B**) LDA algorithm identifies primary research topics of pyroptosis research: “Signal pathways” (red), “Disease-related” (blue), and “Risk and prognosis” (orange). (**C**) A heatmap tracks evolving research focuses over time, emphasising hotspots like signal pathways, inflammasomes, caspase, IL regulation, and apoptosis/ferroptosis. (**D**) The top 10 “MeSH” terms related with pyroptosis highlight metabolism, inflammasomes, and genetics. (**E**) 357 pyroptosis-related genes were identified, showcasing the top 10, with NLRP3, GSDMD, and NLRP1 as leaders.

**Figure 3 cancers-16-00372-f003:**
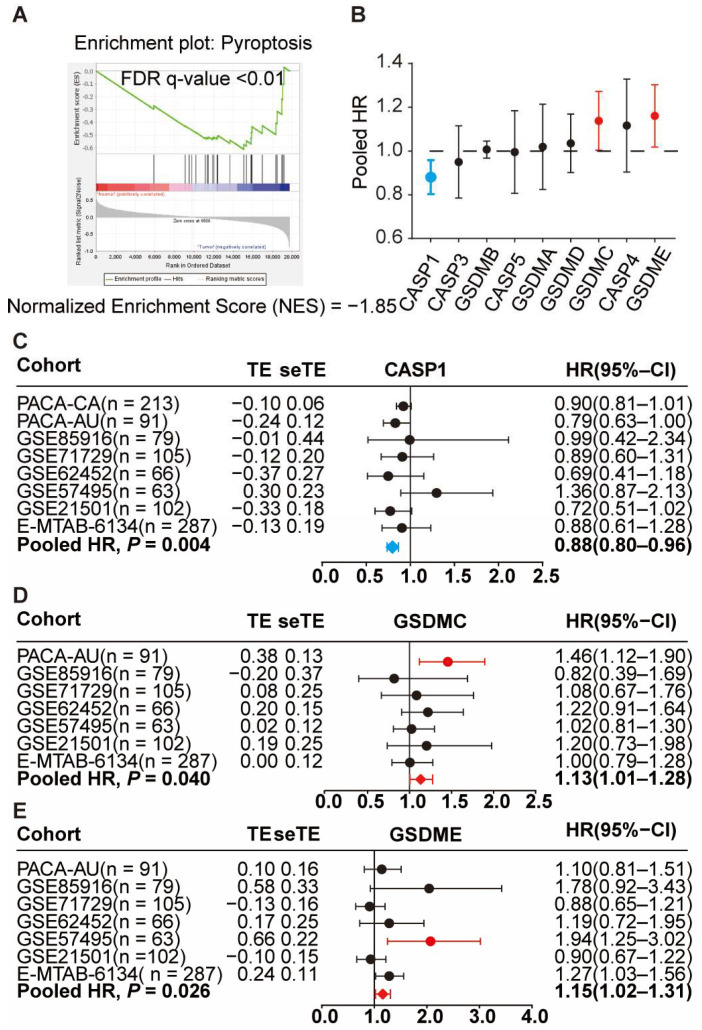
Suppressed pyroptosis pathway and survival correlation in PDAC patients. (**A**) GSEA analysis highlights pyroptosis pathway suppression in PDAC. (**B**) Transcriptome meta-analysis indicates CASP1 expression is inversely linked with OS (HR = 0.88, 95% CI 0.80–0.96, *p* = 0.004), as are GSDMC (HR = 1.13, 95% CI 1.01–1.28, *p* = 0.040) and GSDME (HR = 1.15, 95% CI 1.02–1.31, *p* = 0.026). (**C**–**E**) Details of findings of the meta-analysis. In the representation, red indicates risk factors (HR > 1), while blue signifies protective factors (HR < 1).

**Figure 4 cancers-16-00372-f004:**
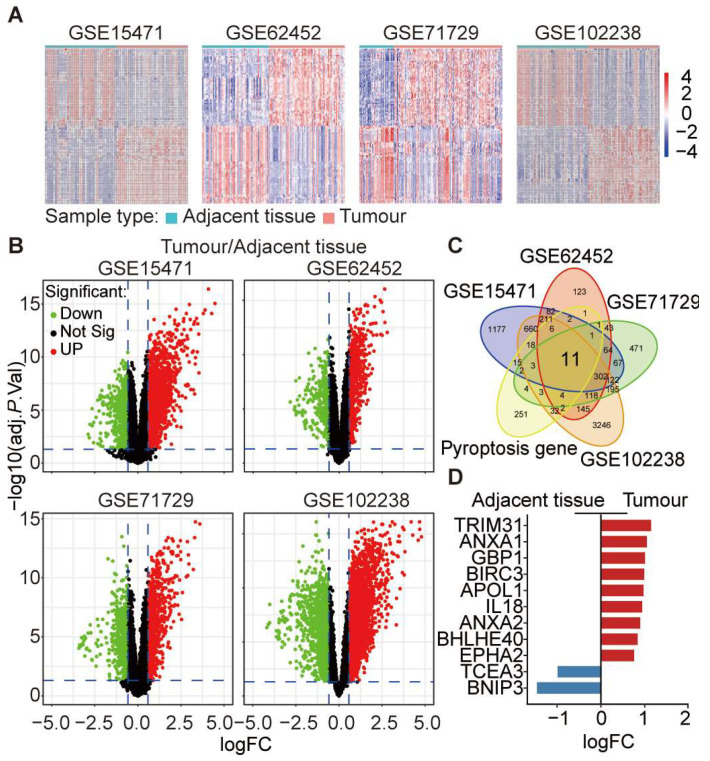
Pyroptosis gene expression variability in PDAC datasets. (**A**) The heat map displays variations in eleven pyroptosis genes across datasets: GSE15471, GSE62452, GSE71729, and GSE102238. (**B**) Volcano plots for each dataset pinpoint significant expression contrasts between PDAC and adjacent non-malignant tissue. (**C**) Intersection analysis reveals differential expression in eleven out of 357 pyroptosis genes. (**D**) Of these, TRIM31, ANXA1, GBP1, BIRC3, APOL1, IL18, ANXA2, BHLHE40, and EPHA2 are upregulated in tumours, whereas TCEA3 and BNIP3 are elevated in adjacent tissue.

**Figure 5 cancers-16-00372-f005:**
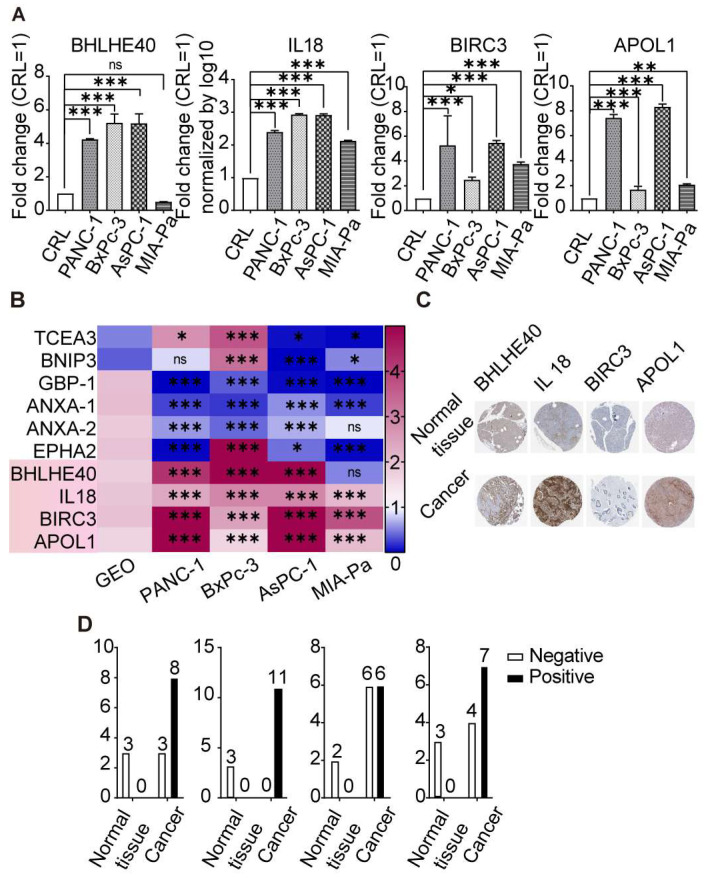
Elevated BHLHE40, IL18, BIRC3, and APOL1 in PDAC tumour samples and cell lines. (**A**) RT-qPCR results for BHLHE40, IL18, BIRC3, and APOL1 in PDAC cell lines MIA-PaCa2, BxPc-3, PANC-1, and AsPC-1 and the non-cancerous pancreatic ductal cell line CRL-4023. (**B**) The heatmap displays expression variations of these genes across the PDAC cell lines according to data extracted from the GEO database. (**C**) According to the Human Atlas Protein database, IHC of BHLHE40, IL18, BIRC3, and APOL1 in normal and PDAC tissue. (**D**) Quantitative presentation of IHC results. Data are graphed as the mean ± SD. ns, not significant; *, *p* < 0.05; **, *p* < 0.01; ***, *p* < 0.001.

**Figure 6 cancers-16-00372-f006:**
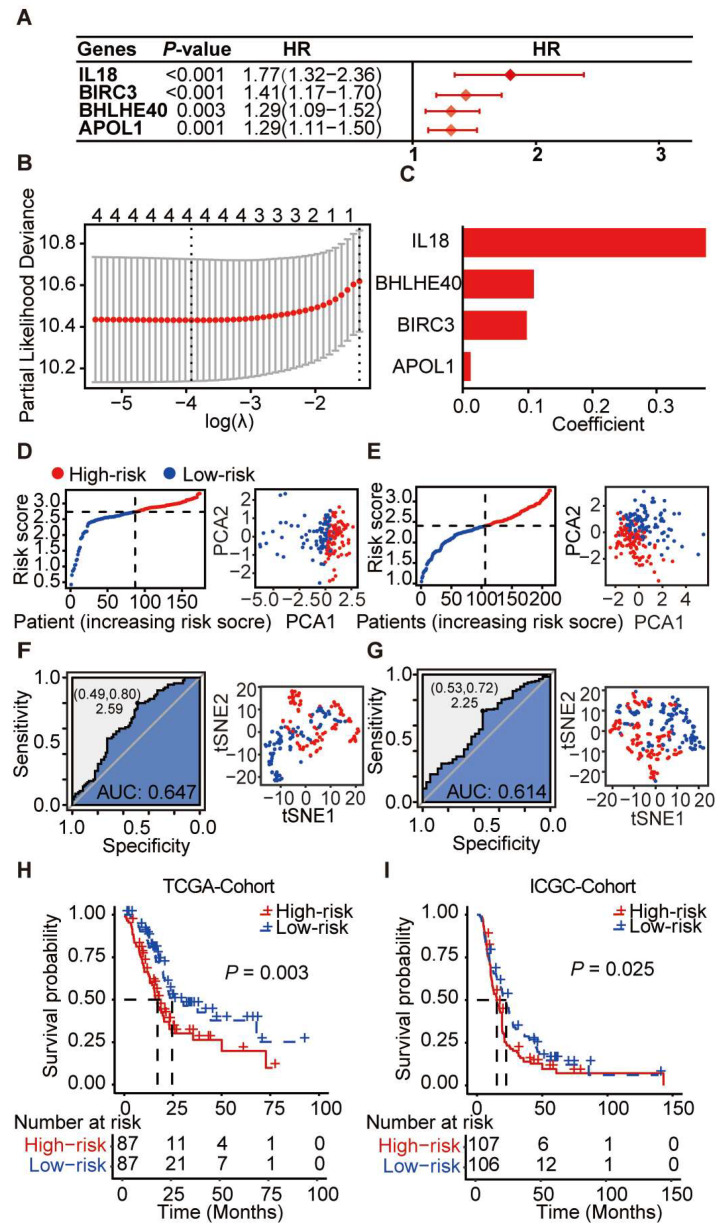
Prognostic index of pyroptosis-related genes in PDAC. (**A**) TCGA’s univariate analysis identified BHLHE40, IL18, BIRC3, and APOL1 as PDAC risk factors with combined expression predicting prognosis. (**B**) LASSO regression optimally selected gene combinations, showing log lambda values. (**C**) Weight histogram for the chosen genes. (**D**) Patient scores from TCGA discerned high from low risk, with principal components analysis (PCA) reinforcing the distinction. (**E**) Similar scoring and PCA for the ICGC database. (**F**) TCGA’s ROC and t-SNE analyses validate and visualise prognosis-based patient clustering. (**G**) Corresponding ROC and t-SNE analyses in ICGC. (**H**) Kaplan–Meier in TCGA and (**I**) ICGC reveals survival differences between scoring groups.

**Figure 7 cancers-16-00372-f007:**
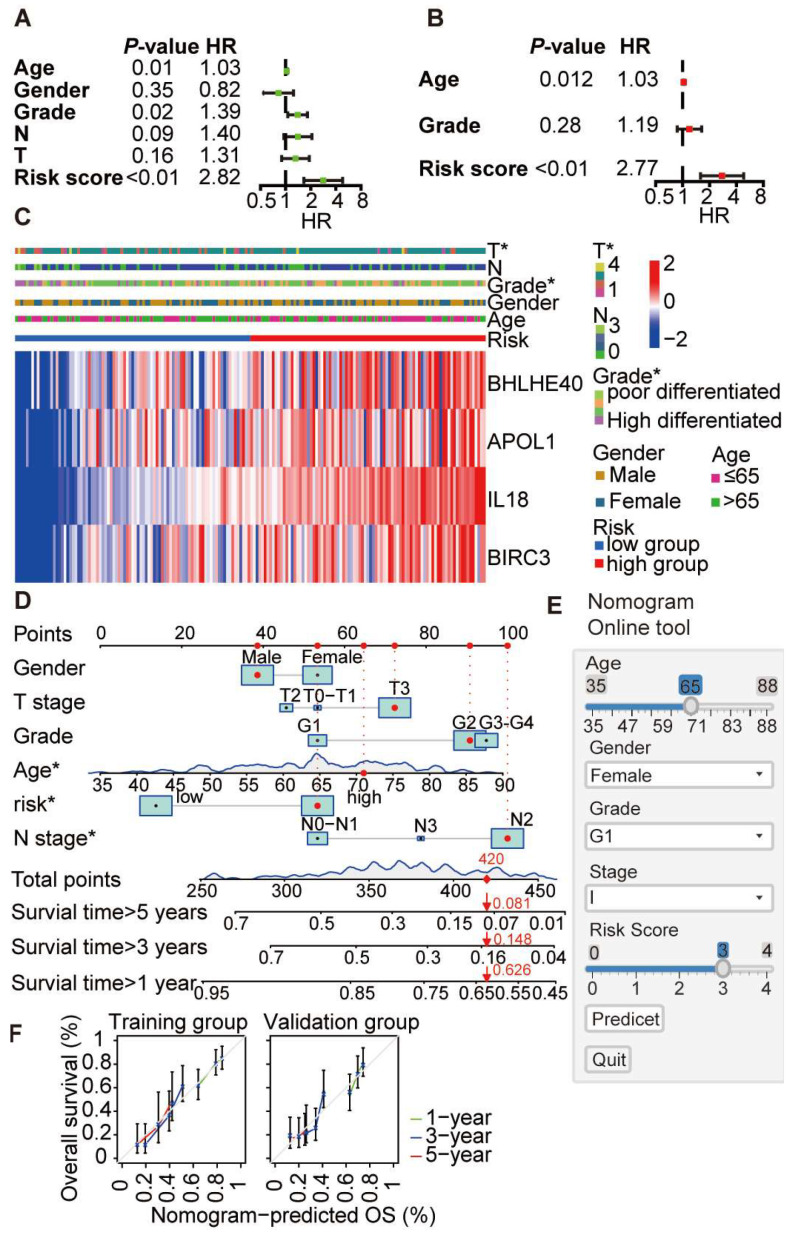
Clinical relevance of the pyroptosis-related prognostic index. (**A**) Univariate analysis showed that score is a significant risk factor for PDAC with HR = 2.82 (1.66–4.82), *p* < 0.001. (**B**) Multivariate analysis confirmed that score is a high-risk factor for PDAC with HR = 2.77 (1.59–4.82), *p* < 0.001. (**C**) Heatmap analysis demonstrates the relationship between the scores and clinical factors. (**D**) A nomogram was constructed using the scoring system and clinical factors. (**E**) The scoring platform/nomogram is accessible at https://nomogram-uniheidelberg.shinyapps.io/DynNomapp/, (accessed on 9 January 2024). (**F**) Calibration curves for the scoring model in TCGA and ICGC. *, *p* < 0.05.

## Data Availability

The datasets can be accessed through the Appendix A and the main body of this manuscript. The code can be accessed through https://github.com/mxdwangdali/2023.12.8PDACpyroptosis, (accessed on 9 January 2024). For additional queries, please contact the corresponding author.

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
