# Peer review of "Multi-Algorithm Analysis Reveals Pyroptosis-Linked Genes as Pancreatic Cancer Biomarkers"

_cancers, 2024, doi:10.3390/cancers16020372_

Round 1
Reviewer 1 Report
Comments and Suggestions for Authors
Dear Authors
The research uses a newly established algorithm to count relevant indicators for the diagnosis of PDCA, and finds that the expression of genes related to pyrotopsis is correlated with patient prognosis. The expression of related genes is confirmed by staining of cell lines and patient specimens. This is an in-depth study. However, there are several issues that require further clarification by the author:
1. The pyroptosis gene expression analyzed by the author is related to which treatment strategies for PDCA, such as which chemotherapy or surgery?
2. Since the author initially targeted pyroptosis as the target gene group, I suggest that the author should add which groups of genes have been reported in predicting the prognosis of PDCA, and should conduct research on the ability to predict different groups of genes.
3. The author uses a new algorithm to predict the prognosis of PDCA. Are there the algorithm tools currently commonly used to predict the prognosis of PDCA or cancer treatment? Can the authors compare the advantages and disadvantages between their tool and others?
Comments on the Quality of English LanguageThis manuscript is good writing and only needs a slight improvement.
Author Response
Dear Reviewer,
Thank you very much for your review of our manuscript. I have attached our detailed responses to each of your comments.
Best regards,
Kangtao

Reviewer 2 Report
Comments and Suggestions for Authors
The Article titled “Multi-Algorithm Analysis Reveals Pyroptosis-Linked Genes as Pancreatic Cancer Biomarkers” by Wang et al, presents and tests a potential novel model for predicting PDAC prognosis. Initially, the Authors state the challenges of PDAC diagnosis at early stages and introduce the concept of the study. Within the study design, the authors analyzed 1,273 PDAV cases and identified 357 pyroptosis-associated genes, including 4-significant genes: BHLHE40, IL18, BIRC3, and APOL1. The Authors suggest that these 4 genes are associated with unfavorable PDAC outcomes, as validated by rt-PCR and predictive modeling experiments on multiple datasets. In the conclusion, the Authors, effectively summarize the key findings, emphasizing the validation of the role of the 4 genes and their significance in PDAC prognosis.
Specific comments:
· The authors provide a comprehensive overview of the challenges associated with PDAC, introduce the concept of pyroptosis and its potential role in PDAC prognosis, and outline the methodology and findings of the study in a clear and scientifically solid language.
· The authors provide a smooth transition between the description of the disease and the introduction of pyroptosis as well as the factors influencing pyroptosis in cancer, such as tumor heterogeneity and biological behaviors and limitations of current diagnostic tests.
· The Authors present coherently and in sufficient detail the three main phases: Discovery, Training, and Validation. In the Discovery Phase, the authors utilize data mining and natural language processing to analyze pyroptosis-related literature from PubMed, aiming to identify key research studies and identify relevant genes associated with pyroptosis. In the Training Phase the authors explain the model based on the upregulation of 4 major pyroptosis-related genes (out of 357). In the Validation Phase they tests the model on new data to assess its accuracy, employing tests like ROC and survival curves.
· The Authors are encouraged to comment more on the relevance of information in public databases due to the concern that in the model, diverse patient populations are included and there may be a possible discrepancy between the criteria and sampling between the sites, as well as the demographic.
· The study reports somewhat inconsistent data, particularly with RT-qPCR. From the discussion, the Authors comment that the discrepancy can be possibly due to the heterogeneity of PDAC cell lines and tumor tissues. The authors are encouraged to comment how this inconsistencies may affect the model.
· The one comment the Authors need to address in more detail is the connection with biology. Understanding the biological pathways and interactions is crucial for developing targeted treatment strategies, and despite the very detailed presentation of the model development (including the details in the supplement material) the article is lacking more details on the biological relevance of the 4 genes.
· The Authors are encouraged to provide more detailed explanation on the nomogram score, more precisely on the basis for the score from the four genes used for evaluating PDAC prognosis.
· The study identifies certain genes, such as BHLHE40, BIRC3, APOL1, and IL18, as having potential clinical relevance, but the specific mechanisms of action of these genes in the context of PDAC are not fully explained. The authors are encouraged to propose further research geared to understanding the detailed mechanisms underlying their roles in pyroptosis and PDAC.
Author Response

(The authors gave the same response as above.)

Reviewer 3 Report
Comments and Suggestions for Authors
This is an interesting and well written paper on an important health-related topic. For the most part, the studies appear to have been conducted carefully, and the authors have used interesting approaches that are worthy of publication. Nonetheless, I have some significant and some minor concerns.
Major:
The authors have chosen to conduct their analysis using data obtained from studies employing microarrays. Due to the limitations of that methodology, the use of microarrays in gene expression analyses has been largely replaced by the use of next generation sequencing methods. It is unclear why the authors have limited their analysis to data obtained using the older methodology, and I saw no discussion of the limitations that the use of that data imposes on their conclusions.
The authors ended up choosing four genes to base their prognostic model on. These genes appear to me to have been chosen because they are differentially expressed between cultured tumor cell lines and a single immortalized control cell line (CRL-4023). However, the cell culture environment and the use of a single immortalized cell line as a control (as well as tumor cells lines for positive controls) may have generated differential gene expression data that has limited correlation with gene expression patterns in human normal and tumor tissues. This in turn likely imposes major limitations on the conclusions that can be reached from these studies, which I did not see addressed. This approach needs to be justified/discussed and/or reevaluated carefully.
The results presented lack important context. While I have no concerns about the validity of their survival analysis of patients relative to expression of the four signature genes, it is unclear if other genes might provide more powerful correlations. Similarly, the authors have not discussed their results in the context of those recently published by Wang et al., 2023 “High pyroptosis activity in pancreatic adenocarcinoma: poor prognosis and oxaliplatin resistance" PMID: 37848674 DOI: 10.1007/s10495-023-01901-w.
The four signature genes appear to me to have weak association with pyroptosis associated publications. TLR2, shown in figure 2, is among the most lowly associated genes, but I find is still associated with 30 pubmed abstracts prior to 2023 (currently 42), while BIRC3, APOL1 and BHLHE40 are currently referenced by 4, 3 and 1 publication, respectively. I think some effort to address whether these genes are actually involved in pyroptosis per se versus any other genes or other cell functions (permutation tests?) would strengthen the study. Additionally, as noted below, access to the code used to generate results might have addressed this concern. The criteria used to accept/reject candidate genes as relevant to pyroptosis are not clear to me.
The authors do not appear to have provided access to their R and python scripts aside from the LDA core code. It is impossible to evaluate the validity of the approaches used (or replicate them) without the code and I think it is increasingly required that authors make these things available upon publication of their work. I suggest adding it to the git repository mentioned in the supplemental data.
Minor issues:
The GSE data sets listed in supplemental data table 1 and the text (section 2.6 and elsewhere) do not entirely match. I also cannot find GSE1022384. This creates some confusion.
The introduction states that “novel biomarkers capable of detecting PDAC at earlier stages” are needed and this is certainly the case. However, the manuscript does not address this issue, but instead identifies prognostic markers. It is not entirely clear that the addition of these four genes to the literature as prognostic markers enhances what is already known about prognostic markers.
It would be helpful to include the numerical data for the graphs shown in figure 2 and elsewhere.
The use of "pyroptosis associated genes" should probably be used throughout, see line 306, for example.
Author Response

(The authors gave the same response as above.)

Round 2
Reviewer 3 Report
Comments and Suggestions for Authors
I feel that the authors have adequately addressed my concerns.